# Very Low-Calorie Ketogenic Diet: A Potential Application in the Treatment of Hypercortisolism Comorbidities

**DOI:** 10.3390/nu14122388

**Published:** 2022-06-09

**Authors:** Valentina Guarnotta, Fabrizio Emanuele, Roberta Amodei, Carla Giordano

**Affiliations:** Dipartimento di Promozione della Salute, Materno-Infantile, Medicina Interna e Specialistica di Eccellenza “G. D’Alessandro” (PROMISE), Sezione di Malattie Endocrine, del Ricambio e della Nutrizione, Università di Palermo, 90127 Palermo, Italy; valentina.guarnotta@unipa.it (V.G.); mnlfrz90s29g273y@studium.unict.it (F.E.); roberta.amodei@gmail.com (R.A.)

**Keywords:** Cushing’s syndrome, glucocorticoid, cortisol, diabetes mellitus, obesity

## Abstract

A very low-calorie ketogenic diet (VLCKD) is characterized by low daily caloric intake (less than 800 kcal/day), low carbohydrate intake (<50 g/day) and normoproteic (1–1.5 g of protein/kg of ideal body weight) contents. It induces a significant weight loss and an improvement in lipid parameters, blood pressure, glycaemic indices and insulin sensitivity in patients with obesity and type 2 diabetes mellitus. Cushing’s syndrome (CS) is characterized by an endogenous or exogenous excess of glucocorticoids and shows many comorbidities including cardiovascular disease, obesity, type 2 diabetes mellitus and lipid disorders. The aim of this speculative review is to provide an overview on nutrition in hypercortisolism and analyse the potential use of a VLCKD for the treatment of CS comorbidities, analysing the molecular mechanisms of ketogenesis.

## 1. Introduction

Cushing’s syndrome is characterized by an exogenous or endogenous excess of glucocorticoids (GCs) resulting in a combination of metabolic disorders, including visceral obesity, type 2 diabetes mellitus, dyslipidaemia and cardiovascular disease. Nutrition has an important role in the management of obesity, diabetes mellitus and cardiovascular disease and may be used as an additional treatment of the metabolic comorbidities in patients with CS waiting to undergo neurosurgery or to reach pharmacologically biochemical remission. 

Traditional methods of weight loss include various types of low-calorie diets calculated on a calorie range of between 800 and 1500 Kcal per day, although calorie requirements vary from individual to individual, so the goal of weight loss can also be achieved with a higher calorie amount [1]. A VLCKD is a nutritional approach characterized by low daily caloric intake (less than 800 kcal/day), low carbohydrate (<50 g/day) and normoproteic (1–1.5 g of protein/kg of ideal body weight) contents [2,3]. This dietogenic protocol leads to the production of ketones, which are then used by other tissues such as the central nervous system, skeletal muscle and heart for energy production [4]. 

A VLCKD has been reported to induce a significant weight loss and improvement in lipid parameters, glycaemic indices and insulin sensitivity, beyond an improvement in neurological and respiratory disorders [5,6,7,8,9,10]. Currently, there are no studies evaluating the effects of a VLCKD and CS; therefore, this area needs further investigation.

Thus, the aim of this speculative review is to focus on the current evidence of nutrition and cortisol levels and the beneficial effects of a VLCKD on metabolic complications and its potential application in patients with hypercortisolism for the treatment of its comorbidities.

## 2. Cushing’s Syndrome

CS is characterized by an excess of GCs that can be exogenous due to a chronic intake of corticosteroids or endogenous owing to the pituitary or adrenal hyperproduction of ACTH or cortisol, respectively. Rarely, the endogenous form can result from an extra-pituitary ACTH-secreting tumour (ectopic CS) [11].

CS is associated with increased mortality and a high risk of cardiovascular disease due to the presence of several comorbidities [12,13,14,15,16]. Comorbidities of CS include metabolic syndrome, characterized by systemic arterial hypertension, visceral obesity, impaired glucose metabolism and dyslipidaemia, polycystic ovary syndrome (PCOS), musculoskeletal disorders, such as myopathy, osteoporosis and skeletal fractures, infections, neuropsychiatric disorders, such as impaired cognitive function, depression or mania, impaired reproductive and sexual function and dermatological manifestations, represented mainly by acne, hirsutism and alopecia [14,15,16,17,18].

The therapeutic approach consists of surgery as the first-line therapy. When patients refuse surgery or it is contraindicated or when a relapse occurs, other therapeutic options including medical therapy, radiotherapy or bilateral adrenalectomy should be evaluated [19,20]. Medical therapy mainly consists of drugs directly inhibiting pituitary ACTH secretion, such as pasireotide and cabergoline, or adrenal steroidogenesis inhibitors such as metyrapone, ketoconazole, osilodrostat, mitotane and etomidate [21,22]. Another medical drug is the glucocorticoid receptor (GR) antagonist mifepristone, which impairs cortisol binding to GR and mainly acts on clinical comorbidities [23]. However, obtaining remission in CS is not always possible and sometimes it is necessary to combine the treatment options. In addition, these patients tend to show the metabolic comorbidities for a long time, sometimes even in the remission phase, and therefore a nutritional approach to improve metabolism should be started. 

### Nutrition and Cushing’s Syndrome

Generally speaking, patients with CS need a low sodium, high-protein and high-calcium diet to prevent muscle and bone loss, respectively. However, an interesting and complex relationship exists between diet macronutrients and cortisol [24]. 

Meal macronutrients have a strong influence on cortisol concentrations, reducing or increasing their levels [25,26,27,28]. Long-standing studies have shown that fasting and starvation are associated with an increase in serum, salivary and urinary free cortisol levels and inadequate suppression of cortisol after a low dose dexamethasone test [29,30,31,32]. However, to what degree caloric restriction cortisol levels increased was not ascertained in any study [33,34]. 

The sympathetic nervous system (SNS) and hypothalamic–pituitary–adrenal axis (HPA) are strictly involved in stress management. An unhealthy behaviour (consumption of highly rich carbohydrate food, chronic stress and reduced sleep) may stimulate cortisol secretion with the development of obesity in subjects who are predisposed for it [35]. By contrast, calorie-restricted diets inducing a decrease in cortisol values are associated with weight loss and reduced chronic inflammation [36].

There are a few clinical studies evaluating the effects of calorie restriction on cortisol levels.

Stimson et al. evaluated the effects of a high-fat/low-carb diet vs. a moderate-fat/moderate-carb diet on cortisol metabolism in obese men [37]. They showed that a lower carb diet was able to regenerate cortisol by increasing the enzyme 11-β-hydroxysteroid dehydrogenase type 1 (11β-HSD1) that activates cortisol and reducing the enzymes (5-alpha and 5-beta reductase) that inactivate cortisol. Other improvements seen in the lower carb group were greater weight loss and improvements in glucose and insulin levels. The regeneration of cortisol in the low-carb group was independent of the difference in caloric intake between the low-carb and the high-carb group, meaning that the number of calories consumed was not a factor in the positive changes seen; rather, it was the carb ratio in the diet that made the difference.

Another study by the same authors showed that dietary macronutrients have different effects on cortisol production [38]. They conducted a study on eight lean men and observed the effects of carbohydrate, high-protein and fat meals on insulin and cortisol levels, showing that all these meals stimulate a rise in cortisol levels, in different ways. Indeed, carbohydrates stimulate both the adrenal cortisol secretion and the extra-adrenal cortisol regeneration mediated by 11β-HSD1, which is present in the liver, adipose tissue and brain, and regenerate cortisol from cortisone releasing it into the bloodstream. By contrast, high-protein and fat meals stimulate adrenal cortisol secretion to a greater degree than extra-adrenal regeneration. The extra-adrenal cortisol regeneration is strictly related to the increase in insulin levels. 

Interestingly, a meta-analysis conducted on 13 studies analysed the effects of fasting, a very low-calorie diet (VLCD) and a low-calorie diet (LCD) on serum cortisol levels [39]. This meta-analysis only included studies that evaluated serum cortisol levels, while it excluded those based on salivary or urinary cortisol levels, in order to avoid heterogeneity of the studies. The results of the meta-analysis showed that short-term calorie restriction was more associated with an increase in cortisol values, compared to a VLCD and LCD, which, in turn, had no long-term effects on serum cortisol values and were less stressful than fasting. In addition, carbohydrate restriction was associated with a decrease in insulin concentration, leading to extra-adrenal cortisol synthesis [39]. 

Furthermore, some amino acids, such as tryptophan, can lead to a decrease in cortisol levels [40]. Similarly, supplementation with phospholipids at the dose of at least 400 mg/day also results in a reduction in cortisol levels [41]. In addition, other nutrients including vitamin B6 and B12, folic acid, lithium, taurine, fermented milk products and sprouts of brown rice, barley and beans, which stimulate the GABAergic system, in turn, can reduce the secretion of CRH, leading to a decrease in cortisol levels [24,41,42,43]. 

## 3. Ketogenic Diet

The ketogenic diet (KD) was used for the first time for the treatment of epilepsy in 1921 [44]. Its use was proposed to mimic the effects of fasting. The KD is a high-fat, low carbohydrate, normocaloric diet. It is characterized by a 4:1 ratio of fat to protein, plus carbohydrates and about 90% of calories are provided by fats (Table 1). In the scenario of the use of a KD, we can identify some variants, the low-calorie ketogenic diet (LCKD) with a calorie intake of 800–1200 Kcal/day and the very-low-calorie ketogenic diet (VLCKD) with a calorie intake of less than 800 Kcal/day (Table 1) [45].

### Biochemistry of Ketogenesis

The process of ketogenesis depends on a reduction in glucose intake and the production of ketone bodies, to boost the mechanisms of the beta-oxidation of fatty acids [46]. The biochemical process takes place in the liver (also to a small extent in the kidney), and becomes very intense in conditions of reduced glucose availability, such as fasting, or in certain pathological conditions, such as diabetes mellitus, due to the lack of antiketogenic activity of the hormone insulin. Metabolism in these circumstances is predominantly dependent on the oxidation of fatty acids from the catabolism of storage lipids.

From a biochemical point of view, the process depends on the inhibition of the glycolytic pathway, which normally guarantees an adequate share of pyruvate and oxaloacetate, used in the Krebs cycle [47]. Under conditions of a lack of glucose, pyruvate and oxaloacetate are used as substrates in hepatic gluconeogenesis to ensure adequate blood glucose levels. By contrast, fatty acids are transported into mitochondria via carnitine palmitoyltransferase (CPT-1), and then broken down into acetyl-coenzyme-A (CoA) by beta-oxidation. The mechanism by which an adequate amount of CoA is ensured is through the reaction between the excess acetyl-CoA molecules that accumulate in the mitochondria, resulting in condensation reactions. Ketogenesis then starts, leading to the formation of ketone bodies: acetoacetate, beta-hydroxybutyrate and acetone [48]. The most important enzyme in this process is beta-hydroxy-methylglutaryl-CoA synthetase. 

The process of ketogenesis is regulated at three levels: free fatty acid production, the hepatic flow of free fatty acids towards esterification or beta-oxidation and the regulation of acetyl-CoA towards oxidation in the Krebs cycle or condensation in ketogenesis. The most important of these steps is regulation of the circulating free fatty acids. These can be regarded as the precursors of ketone bodies. The intensity of ketogenesis is closely related to the proportion of circulating free fatty acids, mainly from adipose tissue. Endocrine and metabolic factors that regulate lipolysis also regulate ketogenesis. Adrenaline, glucagon and pituitary hormones stimulate both lipolysis and ketogenesis, while insulin inhibits the formation of ketone bodies [49]. 

The ketone bodies produced can be used by the heart, muscle and brain for oxidative purposes, to obtain energy [50]. The kidney can also use ketone bodies to a small extent, although in this case they are mainly eliminated in the urine. 

Under healthy conditions, with an adequate intake of carbohydrate-containing foods, the proportion of plasma ketone bodies is 0.3–2 mg/100 mL. If hepatic production exceeds peripheral utilization capacity, a condition of accumulation in the blood (ketosis) occurs, with increased elimination of ketone bodies in the urine (ketonuria). Ketosis, given the acid nature of ketone bodies, can develop into ketoacidosis, one of the most serious complications of type 1 diabetes mellitus [51]. However, ketoacidosis only occurs in conditions of a severe lack of glucose or total insulinopenia since the body has homeostatic pH maintenance activities such as an increased production of NH^4+^ ammonium ions and bicarbonates to prevent the evolution towards acidosis [52]. 

We can define the state of ketosis induced by ketogenesis as ‘controlled’ or physiological ketosis, in which the pH is not altered. Furthermore, in healthy individuals, the increase in circulating ketone bodies stimulates increased insulin secretion, which on the one hand inhibits hepatic ketogenesis and the mobilization of free fatty acids from adipose tissue, and on the other hand promotes the increased elimination of ketone bodies through urine. 

Ketosis is also promoted by a sedentary lifestyle. Exercise leads to the increased muscular production of enzymes (thiophorase and acetoacetyl-CoA thiolase), which promote the uptake of ketone bodies [53]. During fasting, or during a VLCKD, glycaemic levels are maintained in a normal range thanks to the glucostatic function of the liver, both through mobilization from the hepatic glycogen reserve and through the process of gluconeogenesis. Low levels of exogenous glucose lead to a reduction in the amount of insulin produced by the pancreas, and an increase in the production of glucagon, responsible for activating enzyme glycogen phosphorylase, which is essential for the production of glucose from glycogen. In this condition, if peripheral tissues were to use only glucose from the liver, glycogen reserves would be rapidly depleted, so the various organs direct their metabolism towards alternative energy sources, such as ketones. This metabolic shift is caused not only by a reduction in the share of exogenous glucose intake, but also by a reduction in the insulin/glucagon ratio and increased production of adrenaline, which result in the reduced inhibition of lipolysis in adipose tissue. This leads to an increase in the levels of circulating free fatty acids and the enhanced beta-oxidation of fatty acids, which in turn activates ketogenesis [53]. Ketogenesis, whether under conditions of prolonged fasting or glucose deficiency, is a saving mechanism that limits the depletion of glycogen reserves. Some tissues in ketosis mainly use ketone bodies, thus saving glucose and ensuring a reserve for noble, glucose-dependent organs. The sparing effect is not only on glucose metabolism but also on amino acid catabolism since ketosis leads to an inhibition of alpha-keto acid dehydrogenase from the transamination of branched amino acids [54]. The degradation of muscle proteins is therefore counteracted. Triiodothyronine activity is also reduced, leading to a reduction in basal metabolism and protein degradation. A ketogenic diet, however, normally guarantees a protein intake greater than the minimum requirements. 

## 4. VLCKD 

A VLCKD is characterized by, approximately, a 44–43–13% ratio of lipids, proteins and carbohydrates, respectively, and the total energy intake is less than or equal to 800 kcal [55,56,57,58,59,60]. 

In a VLCKD, the process of using ketone bodies as an energy source derived from fatty acids is much more intense, with increased use of these energy sources by tissues such as the heart, kidney, skeletal muscle and central nervous system. Physiologically, acetyl-CoA fuses with oxaloacetate derived from glycolytic processes. Under conditions of slowed glycolysis, such as during a VLCKD, the oxaloacetate produced is preferentially used for neoglucogenetic processes, while the cetyl-CoA molecules derived from the beta-oxidation of fatty acids are used for the production of ketone bodies. The VLCKD is the model with the greatest availability of acetyl-CoA [61,62]. 

A VLCKD plan is normally divided into several phases, with an initial pure ketogenic period of 6–8 weeks (Figure 1). 

Protein preparations containing 18 g of proteins, 4 g of carbohydrates and 3 g of fats may be used initially, but these are gradually discontinued with the introduction of natural protein foods. In phase 1 of the VLCKD, patients are educated to eat high biological value protein preparations five times a day and vegetables with a low glycaemic index. In phase 2, natural proteins including meat/egg/fish are introduced in place of one of the protein preparations at lunch or dinner. In phase 3, natural proteins are introduced in place of the second protein preparation. At the end of the VLCKD, carbohydrates are gradually reintroduced, starting with foods with a lower glycaemic index including fruit and milk products (Phase 4), followed by foods with a moderate glycaemic index such as legumes (Phase 5) and a high glycaemic index (bread, pasta and cereals—Phase 6). This dietetic plan corresponds to an LCD with a daily calorie intake ranging from 1200 to 1500 Kcal/day. At the end of phases 4–6, the patient must be re-educated in order to be able to have a maintenance diet of approximately 1500–2000 Kcal/day and avoid regaining lost weight [58,63].

In a VLCKD, insulin levels are reduced, while glucagon levels increase, and after a few days, circulating levels of free fatty acids and ketone bodies rise. The success of a VLCKD depends not only on the anorectic power of ketone bodies but also on the contribution of certain hormones produced in higher concentrations, such as neuropeptide Y, cholecystokinin and ghrelin. In addition, carbohydrate reduction leads to the rapid consumption of hepatic triglycerides and increased intrahepatic beta-oxidation [64].

### 4.1. Indications and Contraindications of VLCKD

A VLCKD is mainly known for fast weight loss due to the anorectic power of ketone bodies, which reduce the sense of hunger and prevent environmental factors leading to dietary failure. Weight loss can also be optimally achieved by other dietary therapy models, but some indications for the use of the VLCKD include obesity (BMI ≥ 30 kg/m^2^), overweight with metabolic complications (BMI ranging from 25 to 29.9 kg/m^2^) and type 2 diabetes mellitus [65,66,67,68]. Other indications include dyslipidaemia, polycystic ovary syndrome (PCOS), metabolic syndrome and non-endocrine disorders such as asthma, epilepsy, some neurodegenerative disorders, obstructive sleep apnoea syndrome and locomotor system disorders [6,69,70,71,72]. In addition, a VLCKD may be indicated in the pre-operative phase, both for bariatric surgery and for other types of surgery requiring rapid weight loss [73]. In these conditions, adjusting protein requirements promotes weight loss by preventing immune function deficits through protein intake control. 

Finally, a further use of this dietary model concerns competitive sporting activity since an increased protein intake allows muscle mass to be maintained and performance to be improved. 

Equally important are the contraindications to the VLCKD, which currently include type 1 diabetes mellitus, chronic kidney disease, where the protein intake must be adjusted to the renal function parameters, severe hepatic failure due to the predominant beta-oxidation of fatty acids in the liver, cardiac insufficiency, previous stroke, pregnancy and lactation, active neoplasms, severe psychiatric disorders, developmental age and elderly people (relative contraindication) [74,75,76].

### 4.2. Adverse Events of VLCKD

The VLCKD establishes a state of controlled ketosis. This dietary model requires medical monitoring both at the start of the dietary plan, by means of a careful clinical history and assessment of the subject’s state of health, and during the course of the dietary therapy, in order to avoid vitamin deficiencies or electrolyte imbalances that may be associated with the loss of ketone bodies in the urine. This diet may give rise to certain transitory side-effects, such as headaches (due to an increase in circulating ketone bodies), which can be managed by taking analgesic drugs, and halitosis [77]. Other effects are orthostatic hypotension, tachycardia, dehydration, hypoglycaemia, constipation, diarrhoea, nausea, urolithiasis, gallstones, hyperuricemia and muscle weakness [78,79,80,81,82]. The elimination of negatively charged ketone bodies in the urine results in an increased passive loss of positively charged ions such as sodium. This is one of the main reasons why the VLCKD is not suitable in cases of severe heart failure. Supplements of calcium, selenium, zinc, vitamin D and oral alkalis can reduce the incidence of nutritional deficiencies and kidney stones [83]. H2-blockers or proton pump inhibitors may be prescribed to prevent gastrointestinal dysmotility and gastroesophageal reflux [79]. In addition, high-fibre vegetables, sufficient fluids and, if necessary, carbohydrate-free laxatives are recommended to overcome constipation.

### 4.3. VLCKD and Cushing’s Syndrome

Little is known about the effects of a VLCKD on cortisol levels. Indeed, a VLCKD causing a decrease in glucose values may result in an acute increase in cortisol levels to counteract hypoglycaemia stimulating the gluconeogenesis process [83]. However, a prolonged state of ketosis may probably determine a balanced secretion of cortisol, without significant increases. A diet full of high glycaemic index foods can generate a stress condition in the body and favour the deposition of visceral fat and cortisol secretion, leading to a vicious circle. By contrast, a balanced diet has been reported to change body composition and reduce chronic inflammation, acting on cortisol secretion. 

A recent study conducted in 30 obese males showed a significant decrease in salivary cortisol after 8 weeks of a VLCKD compared to the baseline [36]. In addition, these authors suggested that a VLCKD was associated with a significant decrease in serum cortisol by reducing cortisol-binding proteins. 

Although currently the mechanisms associated with a reduction in serum cortisol values during VLCKD are not fully known, we may hypothesize that a VLCKD, by reducing visceral fat, lowers the expression of 11β-HSD1, mainly expressed in visceral adipose tissue, that catalyses the conversion of corticosterone in cortisol and enhances the 11β-HSD2 that inactivates the cortisol in cortisone. 

Currently, there are no clinical studies evaluating the effects of a VLCKD in patients with CS. However, we hypothesize that a VLCKD may be an effective treatment strategy for CS comorbidities including obesity, diabetes mellitus, cardiovascular disease, dyslipidaemia, PCOS and chronic inflammation. Indeed, a VLCKD induces weight loss, reducing visceral fat, by lowering appetite [84] and lipogenesis and increasing lipolysis [85,86]. A VLCKD reduces glucose and insulin levels, improving insulin sensitivity and leading to an improvement in glycaemic control for patients with type 2 diabetes mellitus and an improvement in gonadal function in women (PCOS) [87,88,89,90].

Ketogenesis is associated with a reduction in blood pressure values, which can be attributed in part to the natriuretic effect of ketone bodies [91]. Controlled ketogenesis, which leads to a reduction in excess myocardial lipid content and is also involved in the pharmacodynamic processes of some therapies, is a favourable mechanism for cardiovascular health. In this context, it is important to mention inhibitors of renal sodium-glucose co-transporter type 2 (SGLT2-inhibitors), which, in addition to causing natriuresis and glycosuria, promote a shift towards ketogenesis and a 38% decrease in cardiovascular mortality [92]. The process of ketogenesis has recently been investigated as one of the possible mechanisms associated with the success of SGLT-2 inhibitors on cardiovascular health. Some of these effects appear to be related to the inhibitory action of ketone bodies on the sympathetic nervous system [93]. These effects are metabolic adaptation mechanisms related to the pharmacological induction of glycosuria. In particular, empagliflozin has been shown to promote an increase in endogenous glucose production in response to glycosuria, an increase in lipolysis, fatty acid beta-oxidation and ketogenesis, demonstrated by measuring circulating beta-hydroxy-butyrate levels [94,95]. 

Furthermore, a VLCKD appears to be associated with the reduction in the mediators that promote inflammation, such as TNF-alpha, PAI-1, IL-6, IL-8 and MCP-1, which are also involved in the pathophysiology of cardiovascular diseases and the metabolic syndrome [96,97,98]. 

## 5. Conclusions

It is currently widely known that a VLCKD has beneficial effects on obesity, diabetes mellitus, cardiovascular disease and the improvement of insulin resistance. All these conditions are present in patients with CS. Currently, there are no clinical studies on the use of a VLCKD in patients with hypercortisolism for the treatment of its comorbidities. However, based on the above-mentioned metabolic favourable effects of a VLCKD, it can be hypothesized that it may be successfully employed for the treatment of CS comorbidities (Figure 2). 

To confirm our hypotheses, prospective larger studies on the use of a VLCKD in patients with CS should be performed. 

## Figures and Tables

**Figure 1 nutrients-14-02388-f001:**
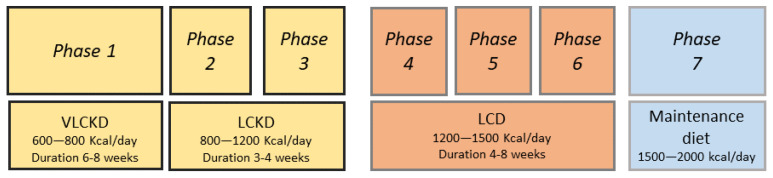
Phases of very low-calorie ketogenic diet (VLCKD) protocol. LCKD: low-calorie ketogenic diet. LCD: low carbohydrate diet.

**Figure 2 nutrients-14-02388-f002:**
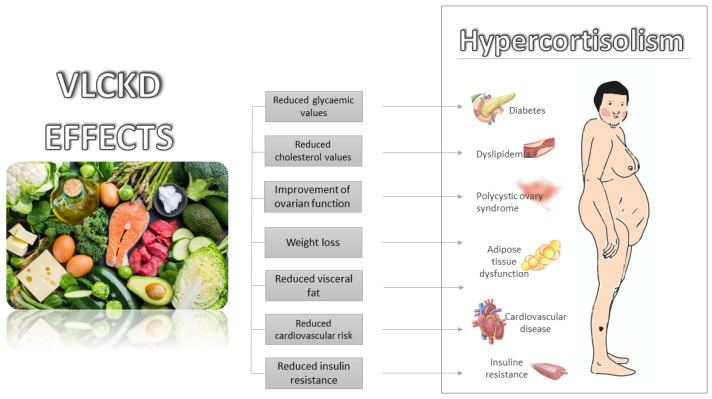
Metabolic effects of VLCKD and potential applications in Cushing’s syndrome comorbidities.

**Table 1 nutrients-14-02388-t001:** Characteristics of classical ketogenic diet (KD), low-calorie ketogenic diet (LCKD) and very low-calorie ketogenic diet (VLCKD).

	KD	LCKD	VLCKD
Caloric intake	Normocaloric	800–1200 Kcal/day	<800 Kcal/day
Carbohydrate (%)	5–10	13	13
Protein (%)	15–20	29	44
Fat (%)	70–80	58	43
Foods	Vegetable oils, fish, eggs, meat, cheese, olives, avocado, coconut	Natural high biological value proteins (1–2 servings) including meat, fish, eggs, processed meat	Replacement meals with high biological value proteins composed by 18 g of proteins, 4 g of carbohydrates and 3 g of fats
Recommendations for use [1]	Epilepsy and resistance to antiepileptic therapy Gliomas and glioblastomas Neurodegenerative diseases (Alzheimer or Parkinson’s disease) Neurocognitive disorders Brain trauma	Obesity BMI 25–35 kg/m^2^ Obesity associated with arterial hypertension or type 2 diabetes mellitus or hypertriglyceridaemia or heart failure or polycystic ovary syndrome Paediatric obesity with epilepsy and/or insulin resistance	Severe obesity Obesity complicated by type 2 diabetes or arterial hypertension or hypertriglyceridaemia or metabolic syndrome or OSAS or arthropathies Obesity with indication of bariatric surgery Adolescents with severe obesity

## Data Availability

Not applicable.

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
