# Peer review of "Very Low-Calorie Ketogenic Diet: A Potential Application in the Treatment of Hypercortisolism Comorbidities"

_nutrients, 2022, doi:10.3390/nu14122388_

Round 1

Reviewer 1 Report

Review of the manuscript entitled: Very low-calorie ketogenic diet: a potential application in the treatment of hypertortisolism comorbidities

There is a contradiction between title (very low-calorie ketogenic diet) and the aim of the paper (ketogenic diet). It should be decided what the paper is on. Very low-calorie ketogenic diet is not the same as ketogenic diet in general.

Paragraph 2, starting from line 43 should be rewritten. E.g. types of ketogenic diets differ by the proportions of macronutrients (fats: carbohydrates+proteins rates by weight) not by the amount and type of dietary fat they contain. Low fat ketogenic diet does not exist. Reference 12 is not properly cited (paper is on low carbohydrate diets not ketogenic diets). It feels that authors do not fully understand the concept of ketogenic diet. Do not mix ketogenic diet and very low calorie ketogenic diet because the biochemistry is totally different. In the same paragraph, no known ketogenic diet (KD) contains 40% of the total energy requirements from proteins (except for VLCKD). Again, you probably mix ketogenic diets (KD) with low-calorie ketogenic diet (VLCKD) but paragraph 3 is devoted to VLCKD. It should be elucidated.

Sentence l.166-167 is unclear (different from what?)

Paragraph 3.2. It should be emphasized that every VLCKD should be carried on under medical not dietary supervision (due to possible severe side-effects)

Minor:

l.87 mithocondria?

l.122 energy utilization of ketone bodies – what do you mean?

l.130 should be ketones instead of ketogenesis

l.307-308 sentence to be deleted.

There are much more minor problems to be thoroughly checked as the ms is careless

Author Response

Review of the manuscript entitled: Very low-calorie ketogenic diet: a potential application in the treatment of hypertortisolism comorbidities

There is a contradiction between title (very low-calorie ketogenic diet) and the aim of the paper (ketogenic diet). It should be decided what the paper is on. Very low-calorie ketogenic diet is not the same as ketogenic diet in general.

Paragraph 2, starting from line 43 should be rewritten. E.g. types of ketogenic diets differ by the proportions of macronutrients (fats: carbohydrates+proteins rates by weight) not by the amount and type of dietary fat they contain. Low fat ketogenic diet does not exist. Reference 12 is not properly cited (paper is on low carbohydrate diets not ketogenic diets). It feels that authors do not fully understand the concept of ketogenic diet. Do not mix ketogenic diet and very low calorie ketogenic diet because the biochemistry is totally different. In the same paragraph, no known ketogenic diet (KD) contains 40% of the total energy requirements from proteins (except for VLCKD). Again, you probably mix ketogenic diets (KD) with low-calorie ketogenic diet (VLCKD) but paragraph 3 is devoted to VLCKD. It should be elucidated.

Thanks for your interesting comment. We focused only on VLCKD, deleting the information on ketogenic diet.

Sentence l.166-167 is unclear (different from what?)

Thanks for the comment. The sentence has been corrected.

Paragraph 3.2. It should be emphasized that every VLCKD should be carried on under medical not dietary supervision (due to possible severe side-effects)

 Thanks for the comment. We changed the sentence as you kindly suggested.

Minor:

l.87 mithocondria?

Thanks for the question. We clarified the process in the text.

l.122 energy utilization of ketone bodies – what do you mean?

Thanks for your kind question. We clarified the sentence in the text.

l.130 should be ketones instead of ketogenesis

Thanks for the comment. We changed it, as suggested.

l.307-308 sentence to be deleted.

Thanks for this useful comment. We deleted it.

There are much more minor problems to be thoroughly checked as the ms is careless

Reviewer 2 Report

Guarnotta et. al., in their review depicted the importance of ketogenic diet in improving various lipid parameters, blood pressure, weight loss etc and its potential use for the treatment of hypercortisolism. While the explanation of various phases of ketogenic diet, the authors focused on the VLCKD and the various case studies depicting its crucial role in treatment of various disorders. I have few comments to make:

  1. Authors are suggested to include more detailed clinical studies/case studies for VLCKD such as cardiovascular system, obesity, PCOS etc. The case studies should include what parameters were included in the studies, the age group, the medical condition, the severity of disease, the difficulties/limitation of the study etc.
  2. Authors are suggested to include the coordination of CNS with HPA since many physiological systems depend on their equilibrium.
  3. I appreciate that the authors provided a comprehensive review on the current understanding of VLCKD. Despite several benefits of VLCKD, there are adverse effects of it. Because of that, the manuscript would be further improved if the author can put their observation in context and how they see the potential of VLCKD in next 5 years will be interesting for the readers.

Author Response

Guarnotta et. al., in their review depicted the importance of ketogenic diet in improving various lipid parameters, blood pressure, weight loss etc and its potential use for the treatment of hypercortisolism. While the explanation of various phases of ketogenic diet, the authors focused on the VLCKD and the various case studies depicting its crucial role in treatment of various disorders. I have few comments to make:

  1. Authors are suggested to include more detailed clinical studies/case studies for VLCKD such as cardiovascular system, obesity, PCOS etc. The case studies should include what parameters were included in the studies, the age group, the medical condition, the severity of disease, the difficulties/limitation of the study etc.

Thanks for this interesting comment. We added more information on the studies reported in the review, as you suggested.

  1. Authors are suggested to include the coordination of CNS with HPA since many physiological systems depend on their equilibrium.

Thanks for the comment. We added it in the text.

  1. I appreciate that the authors provided a comprehensive review on the current understanding of VLCKD. Despite several benefits of VLCKD, there are adverse effects of it. Because of that, the manuscript would be further improved if the author can put their observation in context and how they see the potential of VLCKD in next 5 years will be interesting for the readers.

Thanks for this important suggestion. We added a comment in the conclusions section.

Round 2

Reviewer 1 Report

Abstract contains inconsistencies in definitions of Cushing syndrome

l.14 Cushing’s syndrome is characterized by an endogenous excess of glucocorticosteroids

l.321 Cushing’s syndrome is characterized by an excess of glucocorticosteroids that can be exogenous or endogenous (in addition rarely)

which is true?

l.10 low carbohydrate (<50 g/day)

l.49 a daily carbohydrate quota of 60 grams or less

which is true?

Description of VLCKD is still careless

l.48-52 (about 40% of the total)????  Total what energy, weight???

In one sentence weight and energy quota are mixed up.

Biochemistry of ketogenesis has been considerably improved

l. 143-156 Description of VLKD does not present well defined differences between phases

l.158 circulating levels of free fatty acids and ketone bodies – the sentence lacks a verb.

Part 5. Only last part of very important topic (title topic) but is very poorly written. It should be explained how VLCKD influences cortisol levels. It is not explained.

Last paragraph of the paper l.411-415 is too speculative, was added recently and should be deleted.

Author Response

Abstract contains inconsistencies in definitions of Cushing syndrome

l.14 Cushing’s syndrome is characterized by an endogenous excess of glucocorticosteroids

l.321 Cushing’s syndrome is characterized by an excess of glucocorticosteroids that can be exogenous or endogenous (in addition rarely)

which is true?

Thanks for the comment. We corrected the abstract adding the exogenous excess of glucocorticoids.

l.10 low carbohydrate (<50 g/day)

l.49 a daily carbohydrate quota of 60 grams or less

which is true?

Thanks for the comment. We corrected the sentence in line 49, adding less than 50 grams

Description of VLCKD is still careless

l.48-52 (about 40% of the total)????  Total what energy, weight???

In one sentence weight and energy quota are mixed up.

Thanks for the question. We mean of the total energy intake. We added it in the text.

Biochemistry of ketogenesis has been considerably improved

  1. 143-156 Description of VLKD does not present well defined differences between phases

Thanks for your kind comment. We clarified the differences between the phases 2 and 3.

l.158 circulating levels of free fatty acids and ketone bodies – the sentence lacks a verb.

Thanks for your comment. We added the verb in the text.

Part 5. Only last part of very important topic (title topic) but is very poorly written. It should be explained how VLCKD influences cortisol levels. It is not explained.

Thanks for the comment. As you kindly suggested we tried to hypothesize the possible mechanisms of VLCKD effects on cortisol values. Unfortunately, there are a few studies that assess it.

Last paragraph of the paper l.411-415 is too speculative, was added recently and should be deleted.

Thanks for your suggestion. We deleted the final part of this paragraph.

Reviewer 2 Report

The authors have addressed all the comments, however, the quality of figure 2 is poor and is not clearly visible.. The review can be considered for publishing if they improve the quality of figure 2.

Author Response

The authors have addressed all the comments, however, the quality of figure 2 is poor and is not clearly visible.. The review can be considered for publishing if they improve the quality of figure 2.

Thanks for the comment. We modified the figure 2 improving the quality.

This manuscript is a resubmission of an earlier submission. The following is a list of the peer review reports and author responses from that submission.

Round 1

Reviewer 1 Report

Overview

         In this review, the authors have explained the low-calorie ketogenic diet (VLCKD) and their potential role in hypercortisolism. The author has prepared this review with sufficient information that the information contained therein will be useful to all. This review may be accepted for the publication in this journal. However, I suggest a few comments here to strengthen the review.

  • The author should check for typos in entire manuscript. Example. Line no. 303 VCLKD
  • In VLCKD, it would be even better if details of what foods are suitable for consumption were included.
  • Further, it would be even better to add the recent statistics on the impact of VLCKD on major diseases.

Author Response

 In this review, the authors have explained the low-calorie ketogenic diet (VLCKD) and their potential role in hypercortisolism. The author has prepared this review with sufficient information that the information contained therein will be useful to all. This review may be accepted for the publication in this journal. However, I suggest a few comments here to strengthen the review.

  • The author should check for typos in entire manuscript. Example. Line no. 303 VCLKD

Thanks for the comment. We checked the typing erros and we modified, as you kindly suggested.

  • In VLCKD, it would be even better if details of what foods are suitable for consumption were included.

Thanks for the suggestion. We added the recommended foods for ketogenic diet in the section 2 of the text.

  • Further, it would be even better to add the recent statistics on the impact of VLCKD on major diseases.

Thanks for the suggestion. We updated the references of the section 3

Reviewer 2 Report

Very low-calorie ketogenic diet: an ally in the treatment strategy of hypercortisolism comorbidities

Abstract: definition of very low-calorie ketogenic diet (VLCKD) is not clear. What does hypoglycaemic contents mean? (line 11) What more, in introduction the same abbreviation VLCKD is defined as a very low carbohydrate ketogenic diet (line 28)

What kind of diet do you mean in line 50?

The aim of the review was to study the potential beneficial effects of very low calorie ketogenic diet whereas you allocate very little space to write about it. This review is hugely speculative as there is no published clinical data on this specific subject.

Line 50-51 statement is simply not true as ketogenic diets differ mainly by the amount of carbohydrates but not fat. In fact all ketogenic diets are relatively high in fat. Ketogenic diets differ by the source of carbohydrates (complex sugars vs. simple sugars).

Author Response

Abstract: definition of very low-calorie ketogenic diet (VLCKD) is not clear. What does hypoglycaemic contents mean? (line 11)

What more, in introduction the same abbreviation VLCKD is defined as a very low carbohydrate ketogenic diet (line 28)

Thanks for comments. As you kindly suggested we modified the definition of VLCKD in “low daily caloric intake (less than 800 kcal/day), low carbohydrate (< 50 g/day) and normoproteic (1-1.5 g of protein/kg of ideal body weight) contents”

What kind of diet do you mean in line 50?

Line 50-51 statement is simply not true as ketogenic diets differ mainly by the amount of carbohydrates but not fat. In fact all ketogenic diets are relatively high in fat. Ketogenic diets differ by the source of carbohydrates (complex sugars vs. simple sugars).

Thanks for the comments. We specified the difference in the fat amount between the VLCKD and the LCKD and in the source of carbohydrates as you kindly suggested.

The aim of the review was to study the potential beneficial effects of very low calorie ketogenic diet whereas you allocate very little space to write about it. This review is hugely speculative as there is no published clinical data on this specific subject.

Round 2

Reviewer 2 Report

  1. The aim of the review was to study the potential beneficial effects of very low-calorie ketogenic diet (VLCKD) whereas you allocate very little space to describe this specific kind of the ketogenic diet (e.g the specific composition, indications, biochemistry of the VLCKD).  Authors focus on Cushing's syndrome and on ketogenic diet separately (definition, etiology, risk factors etc.), although do not provide a clear connection between Cushing's syndrome, ketogenic diet and intermediate metabolism.
  2. Introduction does not specify the aim of the study. Authors mix up the very low-calorie ketogenic diet and the very low-carbohydrate ketogenic diet using the same abbreviation (VLCKD) interchangeably, whereas these diets differ in terms of composition, indication, biochemistry, side effects etc (1,4,5).
  3. You describe biochemistry of the ketogenic diet in general but not biochemistry of the VLCKD which is definitely different.
  4. Line 100-101 statement is simply not true as fatty acids are brought into the mitochondria via carnitine palmitoyltransferase (CPT-1) and then broken down into acetyl CoA via beta-oxidation (17).
  5. This review is hugely speculative as there is no published clinical data on this specific subject.

Overall, in the present form the review poorly introduces how and why very low-calorie ketogenic diet influences Cushing's syndrome.